# Cytokines and appetite-regulating hormones in human milk and associations with infant growth across four sites in a longitudinal cohort: The Mothers, Infants and Lactation Quality Study

Sophie Hilario Christensen[1,2*], Jack Ivor Lewis[1], Hanne Frøkiær[3], Peter Riber Johnsen[3], Janet M. Peerson[4], Xiuping Tan[5], Setareh Shahab-Ferdows[4], Daniela Hampel[4,5], Munirul Islam[6], Gilberto Kac[7], Daniela de Barros Mucci[8], Amanda C. Cunha Figueiredo[7], Sophie E. Moore[9,10], Christian Mølgaard[1], Lindsay H. Allen[4,5], Kim F. Michaelsen[1]

**1** Department of Nutrition, Exercise and Sports, University of Copenhagen, Copenhagen, Denmark, **2** Center for Clinical Research and Prevention, Copenhagen University Hospital—Bispebjerg and Frederiksberg Hospital, Copenhagen, Denmark, **3** Department of Veterinary and Animal Science, University of Copenhagen, Copenhagen, Denmark, **4** Western Human Nutrition Research Center, ARS, USDA, Davis, California, United States of America, **5** Department of Nutrition, University of California Davis, Davis, California, United States of America, **6** The International Centre for Diarrhoeal Disease Research, Bangladesh, Dhaka, Bangladesh, **7** The Universidade Federal do Rio de Janeiro, Rio de Janeiro, Brazil, **8** Department of Basic and Experimental Nutrition, Nutrition Institute, Rio de Janeiro State University, Rio de Janeiro, Brazil, **9** Medical Research Council Unit (MRC) The Gambia at the London School of Hygiene and Tropical Medicine, Fajara, The Gambia, **10** Department of Women and Children's Health, King's College London, London, United Kingdom

* sch@nexs.ku.dk

## Abstract

### Introduction

In resource-poor settings, mother-infant dyads are commonly exposed to environmental factors increasing the risk of infectious diseases and possibly influencing the cytokine profile of human milk (HM). Hormones in HM have been proposed to influence appetite-regulation and possibly growth in exclusively breastfed infants.

### Objective

To compare cytokines and appetite-regulating hormone (ARH) concentrations in HM of mothers from four contrasting populations and investigate associations with infant growth.

### Method

HM samples from 825 mothers participating in the Mothers, Infants and Lactation Quality Study from Bangladesh (BD), Brazil (BR), Denmark (DK) and The Gambia (GM) were collected between 1–3.5 months postpartum and analysed for tumour-necrosis factor-α, interferon (IFN)-γ, interleukin (IL)-1β, IL-4, IL-6, IL-8, IL-10, IL-33,

**Data availability statement:** Data cannot be shared publicly due to concerns regarding participant/patient anonymity. All data underlying the findings are to be found in the manuscript. Data requests can be sent to the Special Consultant at (louisekjoelbaek@nexs.ku.dk).

**Funding:** This work is supported, in part, by the Bill & Melinda Gates Foundation (OPP1148405 and INV-002300) and University of Copenhagen. Under the grant conditions of the Bill & Melinda Gates Foundation, a Creative Commons Attribution 4.0 Generic License has already been assigned to the Author Accepted Manuscript version that might arise from this submission. USDA is an equal opportunity employer and provider. The funders had no role in study design, data collection and analysis, decision to publish, or preparation of the manuscript.

**Competing interests:** The authors have declared that no competing interests exist.

**Abbreviations:** ANOVA/ANCOVA, Analysis of variance / Analysis of covariance; ARH, Appetite-regulating hormones; BD, Bangladesh; BMI, Body mass index; BR, Brazil; CF, Complementary feeding/ complementary foods; DK, Denmark; EBF, Exclusive breastfeeding/ Exclusively breastfed; GDM, Gestational diabetes mellitus; GM, The Gambia; HM, Human milk; IFN-γ, Interferon-γ; IL, Interleukin; IQR, Interquartile range; K, Potassium; LAZ, Length-for-Age Z-score; LPS, Lipopolysaccharide; MILQ, Mothers, Infants and Lactation Quality; MUAC, Mid upper-arm circumference; Na, Sodium; ND, Non-detectable; SCM, Subclinical mastitis; SD, Standard deviation; sELISA, Sandwich enzyme-linked immunosorbent assay; Th1/Th2, T helper cells type ½; TNF-α, Tumour-necrosis factor-α; WAZ, Weight-for-Age Z-score; WLZ, Weight-for-Length Z-score; WHO, World Health Organization.

and insulin, leptin and adiponectin. Infant growth was measured twice between 1–5.99 months postpartum. Analysis of covariance was used to compare geometric means of HM markers between the four sites and associations between HM markers and infant growth were investigated using linear regression analysis.

## Results

Differences in geometric means of all HM cytokines and ARHs were found among the four study sites after adjustment for possible explanatory variables. Lowest levels of most HM cytokines were found in BD, whereas highest levels of IFN-γ, IL-4, IL-10 and IL-33 were found in DK. In GM, cytokines and ARHs were inversely associated with weight-for-age and weight-for-length Z-scores.

## Conclusion

We showed significant differences in HM composition of cytokines and ARHs among the four countries. Highest levels of T helper cell type 2 cytokines, which is typically related to increased risk of atopic diseases, were found in DK. The results may reflect the influence of different environmental exposures in the four sites on HM composition, which may be associated with infant growth in GM.

## Introduction

Bioactive compounds such as cytokines and appetite-regulating hormones (ARHs) have been identified in human milk (HM) with maternal factors such as exposure to infection and higher body mass index (BMI) suggested to influence the concentrations [1–3]. In resource poor settings, environmental exposure, *e.g.,* to lipopolysaccharide (LPS) from gram-negative bacteria from poor quality drinking water, sanitation and hygiene, may stimulate the maternal immune system. Studies have demonstrated that HM cells produce both interleukin (IL)-6 and tumour necrosis factor-α (TNF-α) in the presence of LPS, which are produced less in the absence of LPS [4,5]. Thus, the cytokine profile in HM likely reflects the microbial environment of the mother-infant dyads. The cytokines characterize an immune response dominated by T helper cells type 1 (Th1), which may have immunomodulatory effects on the infant's gut, *e.g.,* stimulating epithelial B cell differentiation and inhibiting T cell function [6–9]. Thus, the microbial environment of the mother-infant dyads reflected in the cytokine profile of HM may stimulate the infant immune system in favour of growth. In a previous cohort of 100 mother-infant dyads in urban The Gambia, HM IL-6 and TNF-α were inversely associated with weight-for-age Z-score (WAZ) when assessed cross-sectionally at two-three months [10]. In the absence of appropriate stimuli, the Th1 immune response may fail to be activated, and thus, a Th2 dominating immune response is preserved. A Th2 dominating immune response has been related to an increased risk of atopic diseases in later childhood [11–14], while Th2 cytokines such as IL-4, IL-13 and IL-33 have independently been identified in HM [15].

Furthermore, maternal overweight has been related to higher concentrations of HM cytokines and ARHs including IL-6 and leptin [1,3,16]. These markers are either secreted from the adipose tissue into the circulation and further transported to the mammary glands or secreted from the mammary glands directly [17–19]. A few studies found TNF-α positively correlated with IL-6 in HM [20,21]. This could either reflect the endocrine signalling occurring in plasma, namely secretion of IL-6 from the adipose tissue stimulated by increased TNF-α levels possibly caused by microbial infection, or secretion of both cytokines by the mammary glands, *e.g.,* upon microbial stimuli [22]. A small study from Oklahoma, USA, further reported inverse associations between HM IL-6 and infant weight gain and fat mass, and between HM TNF-α and leptin and infant lean mass and BMI Z-score, respectively, at 1 month postpartum [20]. However, inconsistent results have been reported elsewhere [1,23,24]. Observational studies further report HM leptin and adiponectin associated with WAZ and weight-for-length Z-scores (WLZ) [25,26], lean mass [23], and fat mass [1], yet results appear conflicting. As such, both bacterial exposure and obesity-related inflammation seem to influence HM levels of cytokines and ARHs, which may affect growth. Comparisons of cytokines and ARHs in HM from contrasting populations are limited and adjustment for possible confounding factors is non-standardized, which complexifies the distinguishment of factors contributing to HM composition of cytokines and ARHs.

The objective of the present study was to compare concentrations of cytokines and ARHs in HM from four contrasting populations. We further explored associations between these HM biomarkers and infant growth.

## Methods

### Study design and participants

The Mothers, Infants and Lactation Quality (MILQ) Study [27] recruited pregnant women and their infants from the four study sites Bangladesh (BD), Brazil (BR), Denmark (DK), and The Gambia (GM). Ethical approval for the study was obtained by the study coordinating center; The Institutional Review Board of the University of California, Davis, CA, USA (IRB ID: 920618–1, Protocol HRP-503- MILQ IRB, Department of Health and Human Services FWA No: 00004557). Further ethical approval for conducting the studies in the four respective sites were obtained by the National Ethical Committees in the respective study sites and both oral and written, informed consent were obtained for both mother and infant's participation. The study recruited from 2017 to 2022 due to delays caused by the COVID-19 pandemic (Bangladesh (08-04-2018 to 13-04-2021), Brazil (24-01-2018 to 19-01-2022), Denmark (03-10-2017 to 10-01-2019), The Gambia (03-05-2018 to 20-01-2022)). The examination visits took place either at the study institution, hospital or as home visits. Women were enrolled before entering the third trimester of pregnancy (or later in pregnancy for women in BD and GM) if they had a singleton pregnancy, were between 18 and 40 years old with a height of > 145 cm and with normal- or over-weight (pre-pregnancy BMI of 18.5–29.9 kg/m$^2$ or mid-upper-arm circumference (MUAC) between 21–33 cm in late pregnancy). The women were non-smokers and had a low intake of alcohol (≤5 units per week). Lastly, women were asked to avoid taking multivitamin supplements in the project period except for national recommendations as well as fortified foods (≤3 times a week) except for iodized table salt. Mothers were withdrawn if they developed gestational diabetes (GDM), preeclampsia, and/or anaemia during pregnancy, unless they were willing to consume iron supplements. Full details of sample size calculation, all screening criteria and site-specific protocols are described in the published study protocol [27]. Despite certain site-specific protocols were applied due to cultural or practical aspects [27], standardized protocols for all data in current analyses were applied.

Infants were screened up to two weeks after birth at which the first visit was planned. Infant inclusion criteria included being born at term (gestational week 37–42) with a birth weight between 2500 and 4200 gram (g) and without congenital malformations expected to interfere with infant feeding, growth or development.

Data for the MILQ study were mainly collected at three time periods after birth (Visit 2 (V2) = 1.0–3.49 months, Visit 3 (V3) = 3.5–5.99 months and Visit 4 (V4) = 6.0–8.49 months), and included biological samples, maternal and infant anthropometry, dietary intake and information regarding morbidity, breastfeeding practices and sociodemographic

characteristics. Data in the present analysis included HM samples and maternal anthropometry at V2 and infant anthropometry at V2 and V3. Data from V4 were omitted as complementary feeding (CF) was introduced for most infants at V4, which is likely to influence growth. Mother-infant dyads were further withdrawn if infants had abnormal growth (WAZ, length-for-age (LAZ), and/or WLZ<-2) at V2 and/or V3 and/or were not exclusively breastfed (EBF) at V2. EBF was defined as infants having received only HM aside from the first week, where infant formula was allowed if breastfeeding was not established. We contacted participants who did not attend the visits, and if the families wanted to withdraw, we accepted and asked for a reason. Since inclusion and exclusion criteria as well as examination protocols were standardized across sites, there was no apparent bias related to the study conduction. Data was collected using the Research Electronic Data Capture online system (Vanderbilt University, Tennessee, USA) [28] hosted in GM.

### Milk sample collection and analyses

Non-fasting mature milk samples were collected as full breast expressions using an electric breast pump (Symphony, Medela, Baar, Switzerland). Milk samples were collected at V2 using the opposite breast of that last used for feeding. The breast was cleaned with an alcohol-free wipe or deionized water before sample collection and hereafter emptied in a 250 mL bottle. The milk was gently mixed, and a sample of 30 mL from the total sample was collected under dim light. Any remaining milk was offered to the infant. Samples were aliquoted immediately after collection and stored at -70°C. In DK, samples were transferred to the analysing laboratory (Department of Veterinary and Animal Science, University of Copenhagen, Denmark), where they were thawed and immediately analysed. Samples from BD, BR and GM underwent one extra freeze-thaw cycle as the frozen aliquots from each site were initially shipped to the MILQ laboratory (USDA, ARS, Western Human Nutrition Research Center, California, USA), where they were thawed, re-aliquoted and shipped to the laboratory in DK for analyses.

Whole milk samples (200 µL) were analysed for TNF-α, interferon-γ (IFN-γ), IL-1β, IL-4, IL-6, IL-8, IL-10, IL-33 and the adipokines leptin and insulin using MSD U-plex multiplex immunoassays (Meso Scale Diagnostics, Rockville, USA). Internal standards were run in duplicates on each plate and four samples were run in duplicates on each plate for assessment of inter-assay and intra-assay variability, respectively. Adiponectin was analysed by a sandwich enzyme-linked immunosorbent assay (sELISA) using a human adiponectin duo set (DY1065) from R&D (Biotechne, Minneapolis, MN, USA). Analyses were performed according to manufacturer protocols. HM was diluted 1:10 in PBS with 1% BSA, pH 7.2–7.4 prior to analysis.

The cytokines are representative of proinflammatory cytokines as well as both Th1 an Th2 cytokines reflecting immune activation, while the ARHs are representative of appetite-regulation.

Lower detection limits of cytokines and ARHs were 0.1 pg/mL (TNF-α), 1.24 pg/mL (IFN-γ), 0.15 pg/mL (IL-1β), 0.05 pg/mL (IL-4), 0.22 pg/mL (IL-6), 0.08 pg/mL (IL-8), 0.09 pg/mL (IL-10), 0.30 pg/mL (IL-33), 0.30 pmol/L (insulin), 28 pg/mL (leptin), 0.30 ng/mL (adiponectin). For non-detectable (ND) values, half of the lower detection limit of each HM marker was consistently used for statistical analyses.

As part of quality control of the results, batch variation in analysing kits was identified using two internal standards on all plates. Concentrations were normalized for statistical analyses according to the average concentrations of the applied internal standards by pooling samples that were subsequently aliquoted and stored at -80°C. Furthermore, four samples were assigned in duplicates on each plate to estimate inter- and intra-plate variation, respectively.

Intra assay variability (CV %) were 28 (TNF-α), 15 (IFN-γ), 52 (IL-1β), 43 (IL-4), 22 (IL-6), 13 (IL-8), 25 (IL-10), 34 (IL-33), 9.6 (insulin), 11 (leptin), 11 (adiponectin), whereas inter-assay variability was normalized as described.

Ratios between Th1 (IFN-γ) and Th2 cytokines (IL-4, IL-10 and IL-33) in HM were calculated for each individual and presented as medians and interquartile range (IQR).

Sodium (Na) and potassium (K) were analysed in human milk in order to characterise mothers with subclinical mastitis (SCM) using a molar ratio (Na:K) >0.6 [29,30]. Na and K were analyzed by inductively coupled plasma-mass spectrometry

(ICP-MS) as described elsewhere (Hampel *et al.,* submitted). Briefly, 600 µL milk was digested with 1.5 mL concentrations nitric acid using a MARS 6 microwave digestion system (CEM, Matthews, NC, USA). The digest was diluted to 20 mL before ICP-MS analysis in kinetic energy discrimination (He-KED) mode.

## Anthropometric measurements

Maternal anthropometry included mid-upper-arm circumference (MUAC) measured at screening as well as weight and height measured at V2. Weight was measured using an electronic scale, while height and MUAC was measured to the nearest millimetre (mm) using a stadiometer and a measuring tape, respectively, and using a mean of three measurements. Maternal body mass index (BMI) was calculated by dividing weight in kilograms (kg) by the squared height in meters (m).

Infant anthropometry included weight and length measured at birth, V2 and V3. Weight was measured using an electronic scale to the nearest g and length was measured to the nearest mm using a length board and a mean of three measurements. Both measurements were assessed without diaper, clothes, or accessories on. Weight (kg) and length (cm) were used in combination with age to calculate infant growth Z-scores [31], *i.e.,* WAZ, LAZ and WLZ using the World Health organization (WHO) Anthro software [32]. Growth z-scores were used to exclude participants according to exclusion criteria and to assess associations between HM cytokines and ARHs and infant growth.

All anthropometric measurements were obtained according to standardized procedures and recorded by trained interviewers. Equipment used for anthropometric assessment in each site can be found in Supplementary S1 Table.

## Statistical analyses

Continuous variables are presented as mean ± standard deviation (SD) for normally distributed data and median and IQR for non-normally distributed data. Categorical variables are presented as numbers (*n*) and percentages (%). Normality and homogeneity of variance were checked prior to reporting the final model estimates. Model estimates of the natural log-transformed data were back-transformed when reporting.

Confounding factors and covariates were chosen *a priori* based on current evidence and biologically plausible explanations as the study was exploratory.

Analysis of variance (ANOVA) was initially applied to test whether means of the transformed HM cytokines and ARHs differed between sites, *i.e.,* sites as exposure and cytokines and ARHs as outcomes in separate models. Pairwise comparisons between sites were further tested using Tukey's honest significance test. Additionally, analysis of covariance (ANCOVA) was applied to investigate possible explanatory variables, which included infant sex, infant and maternal age, maternal BMI at V2, parity and SCM. Sensitivity analyses were further conducted stratified for SCM status. For continuous variables, differences between site medians were tested using Kruskal-Wallis test including Dunn's Test for pairwise comparisons with Bonferroni adjustments. For dichotomous variables, difference between sites were tested using the Chi-square test with additional pairwise comparisons with Bonferroni adjustments. Pairwise comparisons were applied without any covariates. Differences in HM cytokine levels between a SCM and non-SCM group were tested using an unpaired t-test. A significance level of 5% was chosen for all statistical tests and analyses.

Change in infant Z-scores from V2 to V3 was explored using linear mixed-effect models with ID as random effect and adjusted for mean-centered age as fixed effects. Secondly, associations between HM cytokines and ARHs and infant growth z-scores were investigated using separate linear regression models for each HM marker at V2 (exposure) and Z-scores at V3 (outcome) adjusted for the respective mean-centered Z-scores at V2 to reduce any influence of regression toward the mean [33]. Additional covariates included infant sex, infant and maternal age, maternal BMI, parity and SCM. Interaction terms between HM marker and site were included to allow for the slopes of the association to vary between sites.

All statistical analyses were conducted as complete case analysis using R software (version 4.1.3; R Foundation for Statistical Computing, Vienna, Austria) [34]. The packages *rstatix* [35], *multcomp* [36], *lsmeans* [37] and *ggplot2* [38] were used for the primary analysis, including visualization. A *p*-value of < 0.05 was considered statistically significant for main effects, whereas a *p*-value of < 0.1 was used for interaction terms.

## Results

A total of 1,932 mother-infant dyads were enrolled in the MILQ study distributed as $n_{BD} = 492$, $n_{BR} = 506$, $n_{DK} = 383$ and $n_{GM} = 551$ (Fig 1). Of these, $n = 825$ were eligible according to inclusion criteria of the present analysis ($n_{BD} = 238$, $n_{BR} = 210$, $n_{DK} = 190$; $n_{GM} = 187$) of which $n = 672$ also attended V3 ($n_{BD} = 238$, $n_{BR} = 152$, $n_{DK} = 177$; $n_{GM} = 105$).

Mothers had a mean age of 27 ± 5 years at inclusion with mothers in BD being the youngest and mothers in DK being the oldest (Table 1). More than two-thirds of mothers in DK were nulliparous at inclusion, whereas ≤42% were nulliparous in BD, BR, and GM. Caesarean deliveries were most common in BD with 57% of all deliveries, whereas caesarean deliveries comprised ≤17% in BR, DK and GM. Birth weight was similar for BD and GM, whereas infants in DK weighed significantly more and were significantly longer than infants from BD, BR and GM (Table 1). A total of 138 mothers (17.4%) were characterised with SCM based on a HM Na:K ratio >0.6 with the lowest and highest prevalence in DK (4.7%) and GM (29.9%), respectively (Table 2).

In linear mixed-effect models, WAZ increased significantly from V2 to V3 for infants in BD and BR ($\beta_{WAZ\text{-}BD} = 0.20$, $p < 0.01$; $\beta_{WAZ\text{-}BR} = 0.08$, $p = 0.04$), while LAZ increased in BD ($\beta_{LAZ\text{-}BD} = 0.21$, $p < 0.01$), DK ($\beta_{LAZ\text{-}DK} = 0.15$, $p < 0.01$) and GM ($\beta_{LAZ\text{-}GM} = 0.16$, $p < 0.01$). WLZ declined from V2 to V3 for infants in GM ($\beta_{WLZ\text{-}GM} = -0.34$, $p < 0.01$), while there was no change in the three other sites (Fig 2).

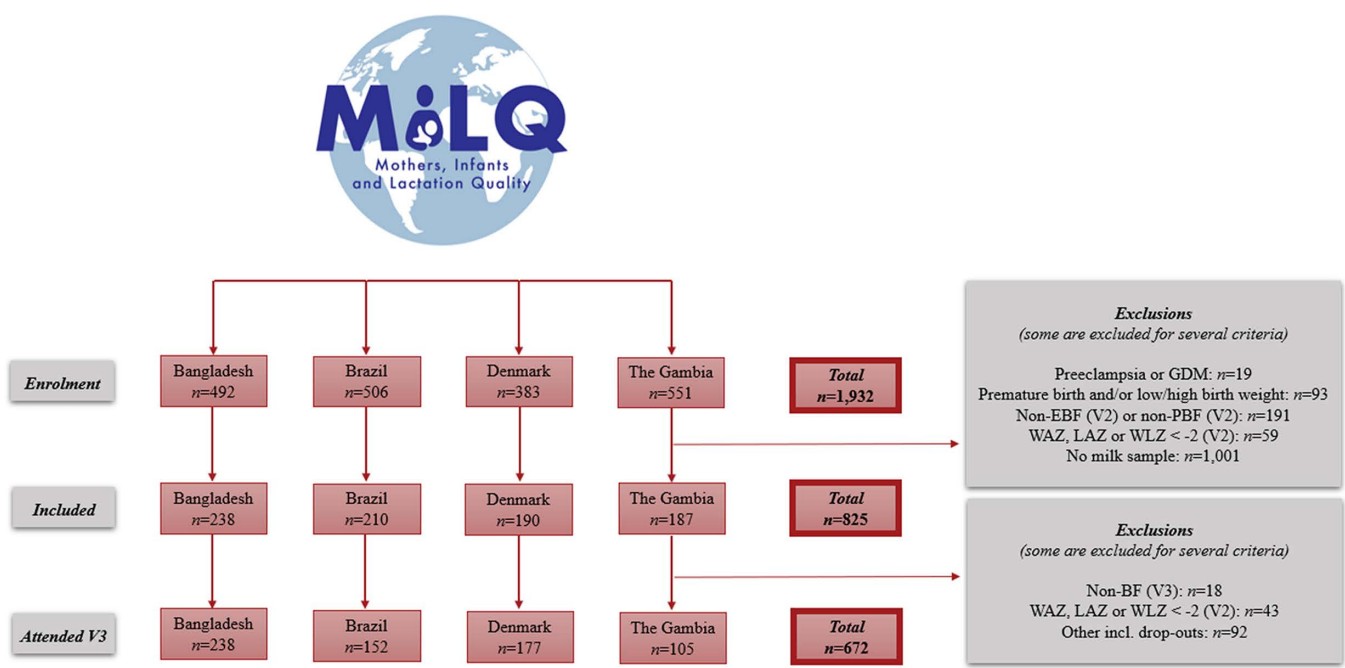

**Fig 1. Flow diagram of participants in the present study.** 1,932 were included in the study of which 1,107 were either excluded, dropped out and/or did not provide a human milk sample. This resulted in 825 included in the present analysis of which 672 also had infant growth measurements available. EBF = Exclusive breastfeeding; GDM = Gestational Diabetes Mellitus; LAZ = Length-for-age Z-score; PBF = Partial breastfeeding; V2 = Visit 2 (1.0-3.49 months); V3 = Visit 3 (3.5-5.99 months); WAZ = Weight-for-age Z-score; WLZ = Weight-for-length Z-score.

**Table 1. Participant characteristics at inclusion and at V2 (1-3.49 months).**

| | Bangladesh (*n*=238) | Brazil (*n*=210) | Denmark (*n*=190) | The Gambia (*n*=187) | All (*n*=825) |
|---|---|---|---|---|---|
| *Maternal characteristics* | | | | | |
| **Maternal age at inclusion** (years) | 24±4[a] | 26±6[b] | 31±3[c] | 28±5[d] | 27±5 |
| **Nulliparous at inclusion** (*n* (%)) | 99 (42)[a] | 88 (42)[a] | 139 (73)[b] | 74 (40)[a,c] | 400 (49) |
| *Mode of delivery* (n(%)) | | | | | |
|   **Vaginal** | 103 (43)[a] | 175 (83)[b] | 178 (94)[c] | 161 (88)[b,c] | 617 (75) |
|   **Caesarean section** | 135 (57) | 35 (17) | 12 (6) | 21 (12) | 203 (25) |
| **Weight, V2** (kg) | 57.4 (8.5)[a] | 64.7 (8.8)[b] | 67.2 (8.5)[b] | 65.5 (10.1)[b] | 63.3 (9.8) |
| **Height, V2** (cm) | 154.9 (3.7)[a] | 160.7 (5.7)[b] | 168.8 (6.1)[c] | 163.9 (5.6)[d] | 161.4 (7.4) |
| **BMI, V2** (kg/m²) | | | | | |
|   *BMI ≥ 25* | 24.0 (3.5)[a] | 25.0 (3.2)[b] | 23.6 (3.0)[a] | 24.5 (3.6)[a,b] | 24.3 (3.4) |
|   *kg/m2* (*n*(%)) | 83 (35)[a,c] | 107 (51)[b] | 54 (28)[c] | 79 (44)[a,b] | 323 (39) |
| *Infant characteristics* | | | | | |
| *Infant sex* | | | | | |
|   **Girls** (*n*(%)) | 118 (50)[a] | 105 (50)[a] | 104 (55)[a] | 93 (50)[a] | 420 (51) |
| **Birth weight** (g) | 3,111±382[a] | 3,322±367[b] | 3,522±369[c] | 3,126±404[a] | 3,265±414 |
| **Birth length** (cm) | 48.2±1.8[b] | 49.1±2.1[a] | 51.8±1.8[c] | 48.9±2.0[a] | 49.5±2.4 |
| *Age* (months) | | | | | |
|   **V2** | 2.6 (0.6)[a] | 2.5 (0.6)[a,b] | 2.0 (0.7)[c] | 2.4 (0.8)[b,d] | 2.4 (0.7) |
|   **V3** | 5.1 (0.6)[b] | 4.9 (0.7)[a] | 4.6 (0.8)[c] | 4.9 (0.9)[a] | 4.9 (0.7) |
| **EBF at V3** (*n*(%)) | 161 (68)[a] | 91 (52)[b] | 104 (59)[a,b] | 108 (86)[c] | 464 (65) |

Data are presented as mean±standard deviation (SD), medians [interquartile range] or counts (percentage (%)). Sites not sharing a letter (a-d) were considered to have statistically different medians and/or distributions at a significance level of 0.05.

EBF=Exclusive breastfeeding; V2=Visit 2 (1.0–3.49 months); V3=Visit 3 (3.5–5.99 months). BMI=Body mass index.

## Human milk composition across study sites

Median concentrations of cytokines and ARHs in HM measured at 1–3.49 months postpartum are presented in Table 2. Concentrations were right-skewed and were normalized by natural log-transformation for the parametric ANOVA test. Differences in geometric means between the four sites were found for all cytokines and ARHs (Fig 3). BD had lower geometric means of all markers except TNF-α, IL-1β and IL-4, whereas BR had higher levels of all three ARHs than the three other sites. DK had higher IFN-γ, IL-4, IL-10 and IL-33 levels than the other sites. Thus, after successful log-transformation of all markers, the significant differences in geometric means between groups correspond to significant differences in the median concentrations of the respective markers between sites.

Significant differences between sites were consistently found for all HM markers except IL-1β when including other explanatory variables in ANCOVA models (Table 3 and Supplementary S2 Fig). An unpaired t-test showed significantly higher levels of all HM cytokines, except from IL-33 and insulin, in the SCM group compared to the non-SCM group (results not shown). Furthermore, significant interactions between site and SCM were found (results not shown), thus, ANCOVA models were further stratified by status of SCM. After stratification, there were still significant differences in levels of all HM cytokines, except for IL-1β, between the four sites both among women with and without SCM (Supplementary S3 Table). IL-1β was significantly different between sites in the SCM group and only borderline significant in the non-SCM group.

Ratios between the Th1 cytokine IFN-γ and the Th2 cytokines IL-4, IL-10 and IL-33 were, respectively, were significantly different between the sites (Table 4). The ratio of IFN-γ:IL-4 was significantly highest in BR and lowest in BD and

**Table 2. Concentrations of human milk cytokines and appetite-regulating hormones measured at V2 (1-3.49 months).**

| HM marker | Bangladesh (n = 238) | Brazil (n = 210) | Denmark (n = 190) | The Gambia (n = 187) | All (n = 825) |
|---|---|---|---|---|---|
| **TNF-α** (pg/mL) | 0.41[a] [0.26; 0.73] ND: 10 (4%) | 0.55[a] [0.30; 1.10] ND: 38 (18%) | 1.14[b] [0.05; 3.02] ND: 55 (29%) | 0.88[b] [0.30; 1.53] ND: 53(28%) | 0.61 [0.30; 1.42] ND: 216 (19%) |
| **IFN-γ** (pg/mL) | 3.63[c] [1.86; 6.40] ND: 12 (5%) | 11.16[a,b] [6.72; 19.39] ND: 0 (0%) | 17.19[a] [6.50; 78.26] ND: 14 (7%) | 9.79[b] [5.18; 18.90] ND: 4 (2%) | 8.17 [3.88; 19.52] ND: 30 (4%) |
| **IL-1β** (pg/mL) | 1.19[a,b] [0.68; 2.37] ND: 3 (1%) | 1.32[a,c] [0.84; 2.42] ND: 0 (0%) | 2.65[d] [0.85; 3.27] ND: 30 (16%) | 1.77[b,c,d] [0.75; 3.42] ND: 0 (0%) | 1.58 [0.74; 3.29] ND: 33 (4%) |
| **IL-4** (pg/mL) | 0.03[a] [0.03; 0.08] ND: 105 (44%) | 0.03[a] [0.03; 0.06] ND: 101 (48%) | 0.24[b] [0.18; 0.50] ND: 64 (34%) | 0.06[c] [0.03; 0.23] ND: 71 (38%) | 0.05 [0.03; 0.18] ND: 341 (41%) |
| **IL-6** (pg/mL) | 0.76[c] [0.31; 2.25] ND: 6 (3%) | 2.06[a] [0.92; 5.02] ND: 0 (0%) | 3.46[b] [1.46; 8.22] ND: 13 (7%) | 2.32[a,b] [1.00; 6.17] ND: 4 (2%) | 1.92 [0.72; 5.09] ND: 23 (3%) |
| **IL-8** (pg/mL) | 85[a] [50; 185] ND: 1 (0.4%) | 196[b] [102; 369] ND: 1 (%) | 190[b] [110; 396] ND: 4 (2%) | 232[b] [99; 508] ND: 0 (0%) | 165 [81; 357] ND: 4 (0.7%) |
| **IL-10** (pg/mL) | 0.04[b] [0.04; 0.18] ND: 89 (37%) | 0.17[a] [0.10; 0.35] ND: 16 (8%) | 0.75[c] [0.30; 1.78] ND: 18 (9%) | 0.20[a] [0.08; 0.40] ND: 23 (12%) | 0.19 [0.05; 0.51] ND: 146 (18%) |
| **IL-33** (pg/mL) | 0.57[a] [0.27; 1.28] ND: 18 (8%) | 5.20[b] [2.16; 13.16] ND: 6 (3%) | 14.16[c] [6.38; 26.71] ND: 6 (3%) | 2.65[d] [1.08; 7.18] ND: 4 (2%) | 3.05 [0.78; 11.57] ND: 34 (4%) |
| **Insulin** (pmol/L) | 109[b] [70; 164] ND: 0 (%) | 206[c] [123; 311] ND: 0 (%) | 145[a] [113; 188] ND: 3 (2%) | 147[a] [91; 253] ND: 0 (0%) | 145 [94; 219] ND: 3 (0.4%) |
| **Leptin** (pg/mL) | 45[b] [17; 114] ND: 16 (7%) | 247[c] [125; 513] ND: 2 (1%) | 126[a] [12; 325] ND: 45 (24%) | 129[a] [56; 303] ND: 14 (7%) | 122 [37; 295] ND: 77 (9%) |
| **Adiponectin** (ng/mL) | 1.46[a] [0.78; 3.25] ND: 10 (4%) | 7.85[b] [4.47; 17.29] ND: 17 (8%) | 2.39[c] [1.54; 3.63] ND: 7 (4%) | 4.05[d] [2.52; 6.69] ND: 4 (2%) | 3.21 [1.52; 6.27] ND: 38 (5%) |
| **Na:K > 0.6** (n(%)) | 40 (16.8)[a] | 33 (15.7)[a] | 9 (4.7)[b] | 56 (29.9)[c] | 138 (17.6) |

Data are presented as medians [interquartile range] or counts (percentage (%)). Sites not sharing a letter (a-d) were considered to have statistically different medians and/or distributions at a significance level of 0.05.

IL = Interleukin; INF-γ = Interferon-γ; K = Potassium; Na = Sodium; ND = Non-detectable; TNF-α = Tumour necrosis factor-α.

DK, compared to the other sites. The ratios of IFN-γ:IL-10 and IFN-γ:IL-33 were both significantly lower in DK compared to the three other sites, and were highest in BR and BD, respectively.

## Associations between human milk composition and infant growth

Higher HM TNF-α, IFN-γ, IL-1β, IL-4 IL-6, IL-8, IL-10 and leptin were associated with lower WAZ in GM, but in none of the other sites (Table 5). Similarly, higher HM TNF-α, IFN-γ, IL-1β, IL-6, IL-8, IL-10, leptin and adiponectin were associated with lower WLZ, while HM insulin was positively associated with LAZ, in GM only. In BR, HM adiponectin was inversely associated with WAZ (Table 5). Neither of the HM markers were associated with infant Z-scores in BD and DK (Table 5).

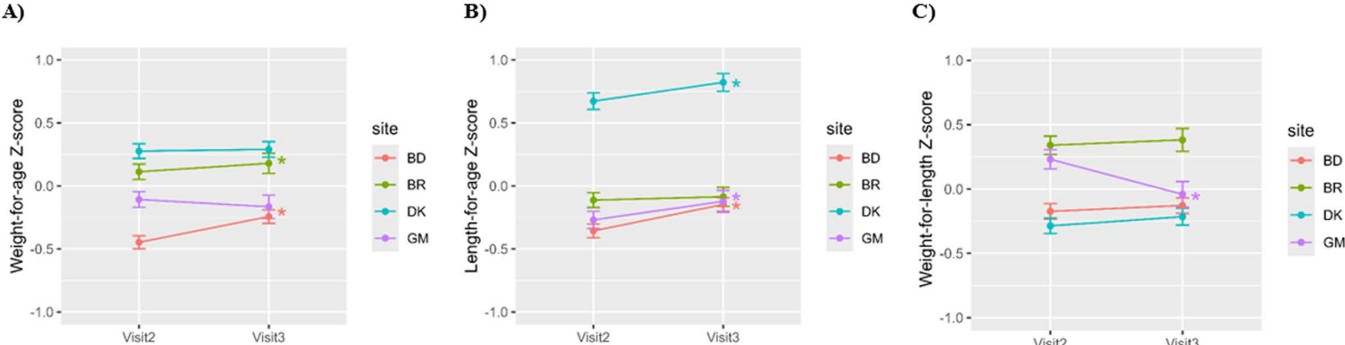

**Fig 2. Mean ± standard error of infant Z-scores across lactation from 1-5.99 months grouped by study site. A)** Weight-for-age Z-score; **B)** Length-for-age Z-score; **C)** Weight-for-length Z-score. Visit 2 = 1-3.49 months; Visit 3 = 3.5-5.99 months. Asterisk (*) indicates significant change from Visit 2 to Visit 3 analysed using linear mixed-effect models with individual as random effect and adjusted for age at visit. BD = Bangladesh; BR = Brazil; DK = Denmark; GM = The Gambia.

## Discussion

In a large sample of well-characterized mother-infant dyads from four contrasting settings, we have shown significant differences in HM concentrations of cytokines and ARHs among the cohorts, *i.e.,* BD, BR, DK and GM. Lower concentrations of most markers were found in BD, while the highest levels of cytokines related to allergy and atopy were found in DK. Adiponectin significantly differed between all four sites, with the highest and lowest concentrations in BR and BD, respectively. Our results further supported an inverse association between HM cytokines and growth, but only in GM.

### Concentration differences in cytokines and appetite-regulating hormones in human milk

HM IL-4, IL-10, and IL-33 levels were significantly higher in DK compared to the three other sites, and lowest in BD (IL-4, IL-10 and IL-33) and BR (IL-4). All three cytokines represent the class of Th2 cytokines, which are normally dominating in the absence of a Th1 immune response, *i.e.,* the "hygiene hypothesis". The hypothesis poses that exposure to bacterial infections or microbial stimuli, *e.g.,* LPS, stimulates the Th1 immune response while inhibiting the Th2 immune response. Conversely, in the absence of a Th1 stimulus, the Th2 response dominates, activating the IgE and eosinophilic responses seen during atopic diseases [14,39]. Thus, the higher levels of IL-4, IL-10 and IL-33 found in DK may reflect the higher prevalence of allergies typically seen in high-income countries (HIC) compared to low- and middle-income countries (LMIC) [40]. The presence of siblings [41] and/or pets [42] in the household as well as household size [43] are factors suggested to contribute to the Th1 stimulated immune response resulting in a less Th2 dominating immune response. The percentage of mothers being nulliparous at inclusion was significantly higher in DK compared to the other sites, yet, adjusting for parity did not change the results. Unfortunately, we did not assess presence of allergy and/or pets in the household, which may partly explain the findings.

A remarkably high prevalence of caesarean deliveries was seen for mothers in BD compared to the three other sites. As our results indicate that the mother's activated immune response is reflected in the cytokine profile in HM, and surgical procedures activate the immune system [44], increased levels of HM pro-inflammatory cytokines could be hypothesized for BD. However, we see the lowest concentrations of IL-6 and IL-8 in BD compared to the other sites. A study from Hungary reported increased plasma levels of IL-6 and IL-8 in women who had caesarean sections compared to vaginal births. However, no differences between the groups were observed on day three after delivery [45]. Furthermore, cytokines have a rather short half-life [46,47], which combined with our findings, support that mode of delivery is unlikely to explain differences in HM cytokines at 1–3.49 months after birth.

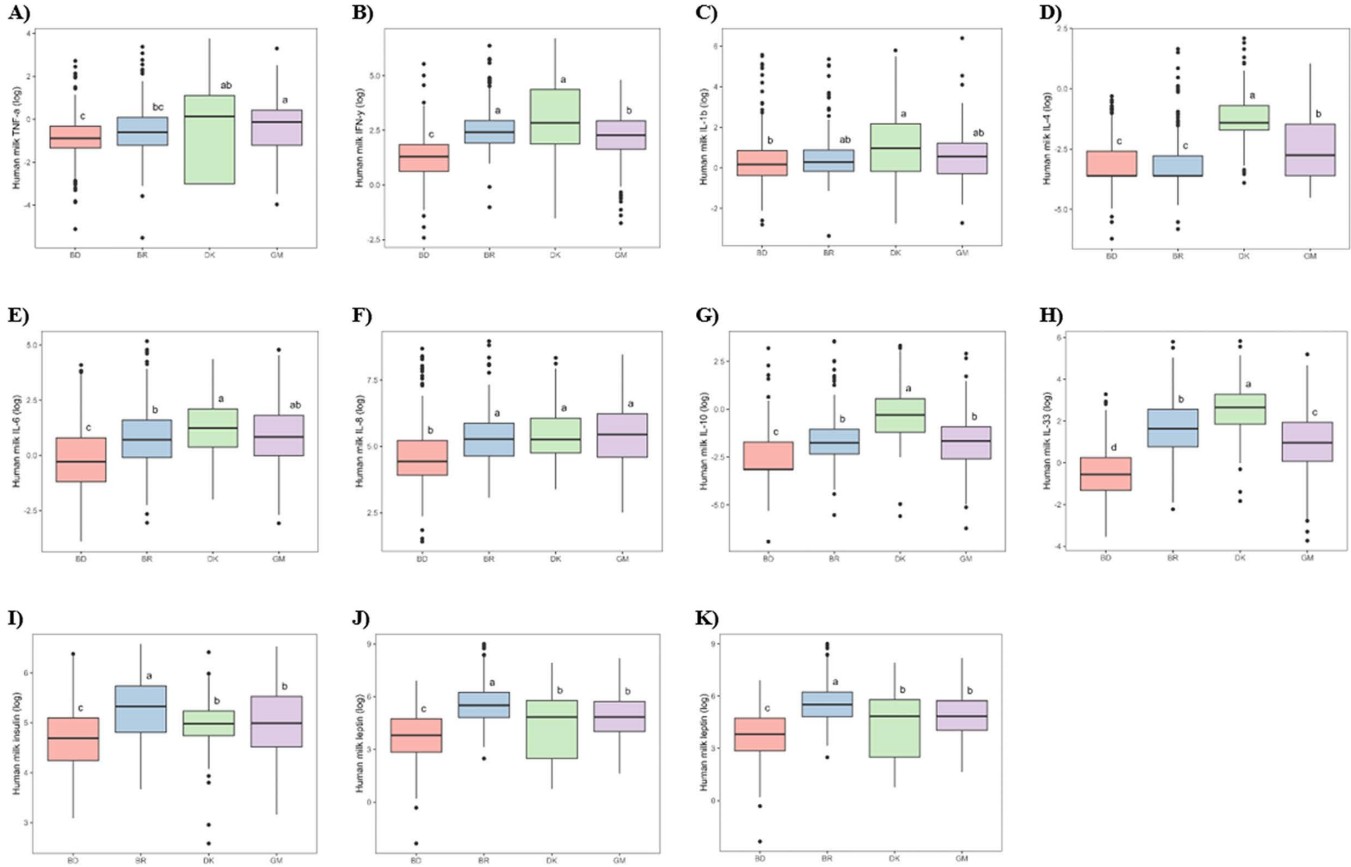

**Fig 3. Boxplots of natural logarithm-transformed human milk cytokines and appetite-regulating hormones at 1.0-3.49 months between the four study sites.** Comparisons were analysed using analysis of variance followed by Tukey's honest significance test for pairwise comparison. Sites not sharing a letter (a-d) was considered statistically different by the pairwise comparison at a significance level of 0.05. BD = Bangladesh; BR = Brazil; DK = Denmark; GM = The Gambia; IFN-γ = Interferon gamma-γ; IL = Interleukin; TNF-α = Tumour necrosis factor-α.

The ratios between the Th1 cytokine IFN-γ and either of the Th2 cytokines IL-4, IL10 or IL-33 have been used in plasma to assess the balance between the activated Th1 and Th2 response with higher ratios reflecting a Th1 dominating response [48,49]. We applied these ratio calculations to our cytokine data in HM, which showed that the ratios were significantly higher in HM from BD and BR and lower in HM from DK. This may support the hypothesis of a Th2 dominating immune response in HM from DK compared to the two other sites. The ratios have previously been applied for HM concentrations of Th1 and Th2 cytokines, where higher ratios have been reported in milk of mothers with SCM compared to without [50]. In the present study, status of SCM was a statistically significant contributor of the differences seen among geometric means of the cytokines and ARHs between the four sites. However, significant differences between sites were found even after stratifying by SCM status indicating that other factors additionally explain the differences. Additionally, Na:K > 0.6 has further been used to describe mammary epithelial permeability [51], which could be caused by differences in milk volume and/or breastfeeding practices, which may also differ between sites in the present study.

Levels of HM insulin, leptin, and adiponectin were highest in BR, where the prevalence of mothers with overweight (BMI ≥ 25 kg/m²) at the time of milk sample collection (V2) was also highest. HM leptin and insulin have been found to correlate with adiposity [1,2,16], likely mediated through circulating levels [52], which may partly explain the findings, although mean BMI values at V2 were within the normal range of weight status in all four sites.

**Table 3. Comparison of covariate-adjusted geometric mean values of human milk cytokines and appetite-regulating hormones across study sites.**

| HM marker | Bangladesh | Brazil | Denmark | The Gambia | F (Dfbetween_sites, Dfwithin_sites); p-value |
|---|---|---|---|---|---|
| | Covariate-adjusted mean [95% CI] | Covariate-adjusted mean [95% CI] | Covariate-adjusted mean [95% CI] | Covariate-adjusted mean [95% CI] | |
| TNF-α (pg/mL) | 0.70 [0.57; 0.85] | 0.86 [0.71; 1.05] | 1.15 [0.89; 1.50] | 0.92 [0.75; 1.14] | $F_{(3,772)} = 3.0^{*}$ |
| IFN-γ (pg/mL) | 5.6 [5.6; 6.8] | 17.8 [14.6; 21.6] | 21.9 [16.9; 28.3] | 9.6 [7.8; 11.8] | $F_{(3,774)} = 36^{***}$ |
| IL-1β (pg/mL) | 2.49 [2.01; 3.09] | 2.53 [2.05; 3.12] | 3.95 [2.99; 5.21] | 2.06 [1.65; 2.57] | $F_{(3,774)} = 5.0^{**}$ |
| IL-4 (pg/mL) | 0.05 [0.05; 0.06] | 0.05 [0.05; 0.06] | 0.40 [0.32; 0.49] | 0.09 [0.08; 0.11] | $F_{(3,774)} = 90^{***}$ |
| IL-6 (pg/mL) | 1.87 [1.54; 2.27] | 4.46 [3.67; 5.40] | 6.82 [5.30; 8.76] | 3.27 [2.68; 4.00] | $F_{(3,772)} = 26^{***}$ |
| IL-8 (pg/mL) | 165 [140; 193] | 313 [266; 368] | 411 [334; 506] | 273 [232; 322] | $F_{(3,763)} = 19^{***}$ |
| IL-10 (pg/mL) | 0.14 [0.11; 0.17] | 0.31 [0.25; 0.39] | 1.04 [0.77; 1.39] | 0.23 [0.18; 0.29] | $F_{(3,771)} = 39^{***}$ |
| IL-33 (pg/mL) | 0.66 [0.53; 0.83] | 5.61 [4.50; 6.99] | 18.17 [13.61; 24.26] | 2.44 [1.94; 3.07] | $F_{(3,773)} = 125^{***}$ |
| Insulin (pmol/L) | 108 [99; 118] | 187 [171; 204] | 163 [145; 183] | 152 [139; 167] | $F_{(3,773)} = 31^{***}$ |
| Leptin (pg/mL) | 54 [44; 65] | 266 [220; 321] | 138 [108; 177] | 141 [116; 172] | $F_{(3,774)} = 60^{***}$ |
| Adiponectin (ng/mL) | 1.40 [1.18; 1.67] | 7.14 [6.02; 8.48] | 2.42 [1.93; 3.03] | 4.19 [3.50; 5.02] | $F_{(3,774)} = 83^{***}$ |

Covariate-adjusted means [95% confidence interval (CI)] of human milk cytokines and appetite-regulating hormones in the four study sites derived from Analysis of Covariance (ANCOVA) models adjusted for infant sex, infant and maternal age, maternal body mass index at V2, parity and subclinical mastitis. Estimates are back-transformed from log-transformed variables used in the ANCOVA. The F-statistics describe between-site variation divided by the within-site variation. P-values <0.05 indicate significant differences in geometric means of the human milk markers between at least one group compared to the others. ' $p < 0.1$;

$^{*}p < 0.05$;

$^{**}p < 0.01$;

$^{***}p < 0.001$.

BD = Bangladesh; BR = Brazil; DK = Denmark; GM = The Gambia; IFN-γ = Interferon gamma-γ; IL = Interleukin; TNF-α = Tumour necrosis factor-α.

In summary, it appears that the included, relevant covariates do not explain the full variation of HM cytokines between the four sites. However, the covariates may partly mediate some of the association, *i.e.,* explain some of the differences found between sites, yet investigating mediating effects were not included in the aim of the present study. In addition, other possible confounding factors such as dietary intake or genetic differences may explain our findings. Gridneva and colleagues further suggests that environmental factors may affect HM composition in combination with other factors, *i.e.,* have a synergistic effect on the HM composition [53]. Examples of HM compounds largely affected by geographical origin include microbiota and lipid composition [54] of which the latter is likely additionally affected by dietary intake, which may also diverge across geographical origins. Regarding immune factors such as cytokines, Ruiz *et al.* observed significant differences in levels of TNF-α, IL-1β and IL-6 in HM from Spain, Sweden, Peru, United States of America, The Gambia, Ghana and Kenya, while levels of IFN-γ were similar across the populations [55]. These results support our findings of geographical differences, yet do not provide further evidence for the explanation hereof.

**Table 4. Ratios between the Th1 cytokine IFN-γ and the Th2 cytokines IL-4, IL-10 and IL-33, respectively.**

| HM marker | Bangladesh (n = 238) | Brazil (n = 210) | Denmark (n = 190) | The Gambia (n = 187) | All (n = 825) |
|---|---|---|---|---|---|
| IFN-γ:IL-4 | 84[a,c] [38; 151] | 322[b] [169; 622] | 62[a] [18; 205] | 108[c] [46; 243] | 114 [42; 315] |
| IFN-γ:IL-10 | 37[a] [17; 80] | 58[b] [32; 117] | 21[c] [11; 74] | 44[a] [20; 107] | 39 [18; 88] |
| IFN-γ:IL-33 | 5.5[a] [2.3; 13] | 2.6[b] [0.8; 8.4] | 1.7[c] [0.5; 3.8] | 3.5[b] [0.9; 14] | 3.1 [1.0; 9.1] |

Ratios are presented as medians [interquartile range]. Sites not sharing a letter (a-d) was considered statistically different by the pairwise comparison at a significance level of 0.05. BD = Bangladesh; BR = Brazil; DK = Denmark; GM = The Gambia; IFN-γ = Interferon gamma-γ; IL = Interleukin.

A higher microbial load may not only affect the cytokine profile in HM, but possibly also increase levels of immunoglobulins, lactoferrin and lysozymes in HM. These are similarly immunoregulatory compounds that are possibly upregulated in HM after microbial exposure [56,57]. The compounds improve the infant gut barrier and thus protect against infections in newborns [58,59]. Varying composition of the HM microbiome has been found between populations previously [55], yet the composition of the above-mentioned immunoregulatory factors have not been systematically investigated across populations before. Additional information about the HM microbiome and protective immune compounds such as immunoglobulin A would add important information to the present study to understand the interplay between microbial exposure and subsequent synthesis of immunoregulatory compounds.

### Associations between cytokines and appetite-regulating hormones in human milk and infant growth

Higher levels of most HM cytokines at 1–3.49 months postpartum were associated with lower WAZ and WLZ at 3.5–5.99 months in GM only. Observations of lower WAZ during the first three months postpartum are in line with the findings by Saso *et al.* from the Gambian cohort [10]. Furthermore, similar observations were recently reported based on pooled findings from 32 and 21 longitudinal cohort studies, respectively [60,61]. It was further concluded, that the incidences of infant stunting [60] and wasting [61] were highest within the first three months postpartum at which time EBF is recommended by the WHO [62–64]. Similar patterns were seen for different geographic regions, including sub-Saharan African countries like GM, but with the greatest prevalence in South Asia [65]. The report further indicates that enteropathogens and intestinal inflammation could explain the findings related to early wasting and stunting due to increased intestinal permeability followed by systemic inflammation, malabsorption, and altered growth hormones [66]. The cytokines in the present study included TNF-α, IFN-γ, IL-1β, IL-6, and IL-8, are all pro-inflammatory cytokines upregulated during microbial exposure. Our findings either indicate that orally ingested cytokines may affect infant growth or that the HM cytokine profile reflects environmental exposures, possibly affecting infant growth. The latter might be the most plausible mechanism explaining our results, although cytokines have been posed to affect the proliferation and differentiation of enterocytes and mucosal barrier in the infant's gut [67,68]. However, neither milk nor gut microbiome, endogenously produced cytokines or growth hormones were investigated in the present study, which would add valuable knowledge to the findings. Possible confounders, such as infant sex or age, have been adjusted for and are therefore less likely to explain current findings. However, uncontrolled confounding, *e.g.,* socio-economic factors, is a possible contributor to the findings. The use of the same strict inclusion and exclusion criteria in all four study sites has likely resulted in a lower variation of socio-economic parameters, which we therefore did not include as potential confounding factors.

Higher HM leptin was further associated with lower WAZ and WLZ in GM. Leptin is suggested to affect secretion of pro-inflammatory cytokines [69], and hence correlations between leptin and the cytokines could be reflected in the

**Table 5. Associations between human milk cytokines and appetite-regulating hormones measured at 1.0-3.49 months and infant Z-scores at 3.5-5.99 months across study sites.**

| HM marker | Bangladesh | | Brazil | | Denmark | | The Gambia | |
|---|---|---|---|---|---|---|---|---|
| | β (95% CI) | p-value | β (95% CI) | p-value | β (95% CI) | p-value | β (95% CI) | p-value |
| *Weight-for-Age Z-score* | | | | | | | | |
| **TNF-α** (pg/mL) | 0.01 (-0.06–0.07) | 0.82 | -0.04 (-0.12–0.04) | 0.28 | -0.02 (-0.06–0.02) | 0.23 | -0.14 (-0.25 – -0.04) | **<0.01** |
| **IFN-γ** (pg/mL) | 0.00 (-0.06–0.06) | 0.97 | 0.03 (-0.06–0.11) | 0.53 | 0.01 (-0.04–0.05) | 0.77 | -0.08 (-0.16 – -0.01) | **0.032** |
| **IL-1β** (pg/mL) | 0.01 (-0.04–0.06) | 0.70 | -0.05 (-0.12–0.03) | 0.23 | -0.01 (-0.04–0.03) | 0.73 | -0.11 (-0.19 – -0.03) | **<0.01** |
| **IL-4** (pg/mL) | 0.00 (-0.06–0.07) | 0.91 | 0.01 (-0.06–0.09) | 0.72 | 0.01 (-0.07–0.09) | 0.85 | -0.10 (-0.18 – -0.01) | **0.028** |
| **IL-6** (pg/mL) | 0.02 (-0.03–0.06) | 0.47 | -0.03 (-0.10–0.04) | 0.41 | -0.04 (-0.09–0.02) | 0.17 | -0.12 (-0.19 – -0.04) | **<0.01** |
| **IL-8** (pg/mL) | -0.02 (-0.08–0.03) | 0.44 | -0.02 (-0.12–0.07) | 0.60 | -0.02 (-0.09–0.06) | 0.63 | -0.15 (-0.24 – -0.06) | **<0.01** |
| **IL-10** (pg/mL) | -0.01 (-0.06–0.04) | 0.77 | -0.02 (-0.09–0.05) | 0.51 | -0.01 (-0.05–0.03) | 0.56 | -0.08 (-0.15 – -0.01) | **0.027** |
| **IL-33** (pg/mL) | 0.05 (-0.00–0.11) | 0.06 | 0.01 (-0.04–0.06) | 0.63 | -0.02 (-0.08–0.04) | 0.50 | -0.01 (-0.08–0.05) | 0.66 |
| **Insulin** (pmol/L) | -0.00 (-0.11–0.11) | 0.95 | -0.07 (-0.20–0.06) | 0.31 | -0.10 (-0.28–0.08) | 0.29 | -0.05 (-0.22–0.12) | 0.57 |
| **Leptin** (pg/mL) | 0.00 (-0.05–0.05) | 0.4 | 0.03 (-0.04–0.10) | 0.40 | -0.02 (-0.07–0.03) | 0.44 | -0.10 (-0.18 – -0.02) | **0.012** |
| **Adiponectin** (ng/mL) | 0.02 (-0.03–0.08) | 0.40 | -0.09 (-0.15 – -0.03) | **<0.01** | 0.01 (-0.08–0.11) | 0.78 | -0.08 (-0.21–0.05) | 0.23 |
| *Length-for-Age Z-score* | | | | | | | | |
| **TNF-α** (pg/mL) | 0.01 (-0.06–0.08) | 0.80 | -0.07 (-0.16–0.02) | 0.12 | -0.00 (-0.05–0.04) | 0.92 | 0.05 (-0.07–0.17) | 0.40 |
| **IFN-γ** (pg/mL) | 0.04 (-0.10–0.03) | 0.26 | 0.05 (-0.05–0.14) | 0.34 | 0.02 (-0.03–0.07) | 0.47 | 0.03 (-0.05–0.12) | 0.46 |
| **IL-1β** (pg/mL) | -0.00 (-0.06–0.06) | 0.94 | -0.04 (-0.13–0.05) | 0.42 | 0.00 (-0.04–0.05) | 0.86 | 0.01 (-0.09–0.10) | 0.88 |
| **IL-4** (pg/mL) | 0.06 (-0.02–0.13) | 0.17 | -0.01 (-0.09–0.08) | 0.87 | -0.01 (-0.10–0.08) | 0.84 | -0.04 (-0.14–0.06) | 0.47 |
| **IL-6** (pg/mL) | -0.00 (-0.06–0.05) | 0.86 | -0.05 (-0.13–0.03) | 0.19 | -0.02 (-0.08–0.04) | 0.50 | -0.02 (-0.10–0.06) | 0.64 |
| **IL-8** (pg/mL) | -0.01 (-0.07–0.05) | 0.77 | -0.03 (-0.13–0.08) | 0.63 | -0.00 (-0.09–0.08) | 0.96 | 0.05 (-0.05–0.15) | 0.35 |
| **IL-10** (pg/mL) | -0.01 (-0.07–0.04) | 0.62 | -0.03 (-0.11–0.04) | 0.40 | -0.00 (-0.05–0.04) | 0.90 | -0.02 (-0.10–0.06) | 0.65 |
| **IL-33** (pg/mL) | 0.05 (-0.01–0.11) | 0.11 | 0.00 (-0.05–0.06) | 0.92 | -0.04 (-0.11–0.03) | 0.22 | 0.05 (-0.02–0.13) | 0.16 |
| **Insulin** (pmol/L) | -0.08 (-0.20–0.05) | 0.21 | 0.02 (-0.13–0.17) | 0.79 | -0.18 (-0.38–0.03) | 0.10 | 0.22 (0.03–0.42) | **0.026** |
| **Leptin** (pg/mL) | -0.02 (-0.08–0.04) | 0.43 | 0.02 (-0.06–0.10) | 0.62 | -0.03 (-0.08–0.03) | 0.37 | 0.01 (-0.08–0.10) | 0.88 |
| **Adiponectin** (ng/mL) | -0.00 (-0.06–0.06) | 0.93 | -0.05 (-0.13–0.02) | 0.15 | 0.04 (-0.07–0.15) | 0.46 | -0.01 (-0.16–0.13) | 0.86 |

**Table 5.** (Continued)

| HM marker | Bangladesh | | Brazil | | Denmark | | The Gambia | |
|---|---|---|---|---|---|---|---|---|
| | β (95% CI) | p-value | β (95% CI) | p-value | β (95% CI) | p-value | β (95% CI) | p-value |
| *Weight-for-Length Z-score* | | | | | | | | |
| **TNF-α** (pg/mL) | -0.00 (-0.09–0.08) | 0.91 | 0.02 (-0.09–0.12) | 0.78 | -0.03 (-0.08–0.02) | 0.25 | -0.27 (-0.41 – -0.13) | **<0.001** |
| **IFN-γ** (pg/mL) | 0.01 (-0.07–0.09) | 0.84 | -0.01 (-0.12–0.11) | 0.93 | 0.01 (-0.04–0.07) | 0.67 | -0.14 (-0.24 – -0.04) | **<0.01** |
| **IL-1β** (pg/mL) | -0.00 (-0.07–0.07) | 0.99 | -0.06 (-0.17–0.04) | 0.23 | 0.00 (-0.05–0.05) | 0.94 | -0.17 (-0.28 – -0.06) | **<0.01** |
| **IL-4** (pg/mL) | -0.07 (-0.16–0.03) | 0.17 | 0.03 (-0.07–0.14) | 0.53 | -0.01 (-0.12–0.10) | 0.87 | -0.11 (-0.22–0.01) | 0.077 |
| **IL-6** (pg/mL) | 0.01 (-0.05–0.08) | 0.66 | -0.00 (-0.09–0.09) | 0.99 | -0.02 (-0.09–0.06) | 0.66 | -0.15 (-0.25 – -0.05) | **<0.01** |
| **IL-8** (pg/mL) | -0.04 (-0.11–0.04) | 0.34 | -0.04 (-0.16–0.09) | 0.56 | -0.02 (-0.12–0.08) | 0.72 | -0.25 (-0.37 – -0.13) | **<0.001** |
| **IL-10** (pg/mL) | 0.01 (-0.06–0.07) | 0.84 | -0.01 (-0.10–0.09) | 0.91 | -0.01 (-0.06–0.05) | 0.85 | -0.10 (-0.20 – -0.01) | **0.029** |
| **IL-33** (pg/mL) | 0.01 (-0.06–0.08) | 0.81 | 0.02 (-0.05–0.09) | 0.59 | -0.00 (-0.08–0.08) | 0.93 | -0.08 (-0.17–0.01) | 0.069 |
| **Insulin** (pmol/L) | 0.02 (-0.13–0.17) | 0.76 | -0.16 (-0.34–0.01) | 0.07 | -0.03 (-0.28–0.22) | 0.82 | -0.23 (-0.46–0.00) | 0.055 |
| **Leptin** (pg/mL) | 0.01 (-0.06–0.08) | 0.87 | 0.01 (-0.08–0.10) | 0.88 | -0.00 (-0.07–0.06) | 0.92 | -0.11 (-0.22 – -0.01) | **0.036** |
| **Adiponectin** (ng/mL) | 0.03 (-0.04–0.10) | 0.41 | -0.08 (-0.16–0.01) | 0.08 | -0.01 (-0.13–0.12) | 0.94 | -0.18 (-0.35 – -0.00) | **0.045** |

Linear regression models of separate logarithm-transformed human milk marker and infant growth Z-scores included an interaction term between HM marker and site and were further adjusted for infant sex and mean-centered age, maternal age and maternal body mass index at V2, parity and SCM as additive covariates. Sample size varied between 645–653 depending on data availability.

BD = Bangladesh; BR = Brazil; DK = Denmark; GM = The Gambia; IFN-γ = Interferon gamma-γ; IL = Interleukin; LAZ = Length-for-age Z-score; TNF-α = Tumour necrosis factor-α; V2 = Visit 2 (1.0–3.49 months postpartum); V3 = Visit 3 (3.5–5.99 months postpartum); WAZ = Weight-for-age Z-score; WLZ = Weight-for-length Z-score.

findings. Higher levels of HM adiponectin were additionally associated with lower WAZ in BR only, representing either finding by chance or possibly indicating similar reflections of environmental exposures through HM.

Importantly, infants in the present cohort were excluded if they were either stunted (LAZ<-2), wasted (WLZ<-2) or underweight (WAZ<-2) at any time point, thus changes in Z-scores likely reflect the normal variation in infant growth in these populations. The mothers in the present cohort were financially capable of consuming a varied diet. In summary, our findings indicate a trend towards higher HM cytokines and declining infant Z-scores possibly reflecting environmental factors influencing infant growth. This could be more pronounced in populations with socioeconomic challenges. Finally, the findings suggest important associations, but do not confirm causal inferences.

## Strengths and limitations

This is the first study to compare cytokines and ARHs in HM across contrasting populations using standardized methodologies, in particular the method of HM sample collection and infant growth assessment. Despite the analyses are secondary to the MILQ study, the sample size of 825 dyads is rather large compared to current literature [1,55]. The relatively narrow confidence intervals provided in, *e.g.,* Tables 3 and 5 indicate that our results present with a high level of certainty. As such,

the large group of 825 mother-infant dyads enabled us to uniquely investigate differences between populations within a group of apparently healthy dyads enrolled using the same eligibility criteria and same standardized milk sampling protocol.

One of the study's main limitations includes using assays that are not yet validated, and are currently unavailable, which may have influenced the estimation of concentrations. Especially as most cytokines were found in lower concentrations than reported elsewhere [1,10,25,70–72], systematic underestimations of concentrations may have occurred. However, compared to the findings by Saso *et al.* [10], concentrations of IFN-γ, IL-1β, IL-4, IL-6 and IL-10 were higher in the present study, while TNF-α concentrations were generally lower. Mothers in the present analysis and the study by Saso *et al.* were all recruited from a similar urban area in The Gambia, and thus differences between environmental factors are unlikely to explain the observed differences. COVID-19 occurred during data collection in the present study, which may explain the higher concentrations if the SARS-CoV-2 virus can activate the maternal immune response accordingly. In addition, DK was the only site to complete data collection before COVID-19, which may partly explain the higher concentrations of most cytokines in DK compared to the three other sites. However, possible explanations regarding COVID-19 are speculative. Furthermore, internal standards applied on all plates enabled normalization of concentrations and thus, unvalidated assays should constitute a minor concern for the overall results. In summary, despite systematic underestimation may have occurred, relative comparisons between sites are likely unaffected.

Samples from DK were analysed 1.5 years before the analysis of BD, BR and GM samples, which additionally underwent one extra freeze-thaw cycle. In that regard, inter-assay variation was assessed using internal standards and concentrations were normalized accordingly. Some intra-assay CV% were relatively high, *e.g.,* for IL-1β, which reflects generally low concentrations. Finally, ≤ 42%, ≤ 27%, and ≤ 30% of all samples were below the detection limit for IL-4, TNF-α, and IFN-γ, respectively, with site variations. As these are probably valid results, half the lower concentration was used in analyses. Finally, the high number of analyses represents an increased risk of results found by chance, which may be seen especially for associations between HM cytokines and infant growth. To summarize, these factors likely affected estimates of HM concentrations, especially when comparing concentrations between sites. However, the methods were considered appropriate for the overall exploration of comparisons and investigating associations with infant growth.

Lastly, the participants represent a selected group due to strict inclusion and exclusion criteria related to the primary aim of the MILQ study. Thus, there is a risk of selection bias affecting the present findings, which may limit the external validity. Despite an extensive data collection and adjustments for known possible confounding factors, the risk of unmeasured confounding remains. The mothers were in general healthy with uncomplicated pregnancies and infants were born healthy and grew normally within the first 8.5 months. Yet, the populations represent contrasting contexts and ethnic and environmental differences appear relevant to investigate in the present cohort.

## Conclusion

In conclusion, we found significant differences in HM concentrations of cytokines between mothers from Bangladesh, Brazil, Denmark, and The Gambia. Th2 cytokines typically related to atopic diseases dominated HM from Denmark, while concentrations of most cytokines were lowest in BD. Furthermore, higher levels of the majority of HM cytokines were associated with lower weight-for-age and weight-for-length Z-scores of infants in The Gambia.

Our results may support that maternal and environmental exposure differences between contexts are likely important for determining the HM concentrations of these bioactive compounds.

However, infants included in the present study were neither wasted, stunted nor underweight, indicating that variations within HM composition of cytokines and ARHs affected growth to a slighter degree. HM cytokines may still have growth regulatory effects, although not to the extent seen by wasting and stunting. In settings with substantial socioeconomic inequality, HM cytokines may have a more pronounced influence on infant growth, likely in combination with other environmental, dietary and socioeconomic factors.

Our findings enhance the understanding of important factors affecting HM composition, while indicating a possible association with infant growth. Understanding the complex interaction between mother and infant through HM may be key in promotion of healthy early infant growth and development especially in resource-poor settings. Future studies would benefit from disentangling the factors affecting HM composition, which requires both a comprehensive and longitudinal data collection.

## Supporting information

**S1 Table. Equipment used for assessment of maternal and infant anthropometry in the respective study sites.** (TIF)

**S2 Fig. Pairwise comparison of geometric means of human milk cytokines and appetite-regulating hormones at 1.0–3.49 months between sites using Analysis of Covariance and Tukey's honest significance test for pairwise comparisons.** BD = Bangladesh; BR = Brazil; DK = Denmark; GM = The Gambia; IFN-γ = Interferon gamma-γ; IL = Interleukin; TNF-α = Tumour necrosis factor-α.
(TIF)

**S3 Table. Comparison of covariate-adjusted geometric mean values of human milk cytokines and appetite-regulating hormones across study sites and stratified by status of subclinical mastitis.**
(TIF)

## Acknowledgments

We thank all the mothers and infants participating in the MILQ study as well as all employers of the MILQ team.

## Author contributions

**Conceptualization:** Sophie Hilario Christensen, Jack Ivor Lewis, Munirul Islam, Gilberto Kac, Sophie E. Moore, Christian Mølgaard, Lindsay H. Allen, Kim F. Michaelsen.

**Data curation:** Jack Ivor Lewis, Xiuping Tan, Setareh Shahab-Ferdows, Daniela Hampel, Munirul Islam, Gilberto Kac, Daniela de Barros Mucci, Amanda C. Cunha Figueiredo, Sophie E. Moore, Christian Mølgaard, Kim F. Michaelsen.

**Formal analysis:** Sophie Hilario Christensen, Hanne Frøkiær, Peter Riber Johnsen.

**Funding acquisition:** Lindsay H. Allen.

**Methodology:** Sophie Hilario Christensen, Jack Ivor Lewis, Hanne Frøkiær, Janet M. Peerson.

**Project administration:** Sophie Hilario Christensen.

**Visualization:** Sophie Hilario Christensen.

**Writing – original draft:** Sophie Hilario Christensen.

**Writing – review & editing:** Jack Ivor Lewis, Hanne Frøkiær, Peter Riber Johnsen, Janet M. Peerson, Xiuping Tan, Setareh Shahab-Ferdows, Daniela Hampel, Munirul Islam, Gilberto Kac, Daniela de Barros Mucci, Amanda C. Cunha Figueiredo, Sophie E. Moore, Christian Mølgaard, Lindsay H. Allen, Kim F. Michaelsen.

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
