## [Decision Letter · Decision Letter 0]

6 Jan 2025

PONE-D-24-37315Cytokines and appetite-regulating hormones in human milk and associations with infant growth across four sites in a longitudinal cohort: The Mothers, Infants and Lactation Quality StudyPLOS ONE

Dear Dr. Christensen,

Thank you for submitting your manuscript to PLOS ONE. After careful consideration, we feel that it has merit but does not fully meet PLOS ONE’s publication criteria as it currently stands. Therefore, we invite you to submit a revised version of the manuscript that addresses the points raised during the review process. Please submit your revised manuscript by Feb 20 2025 11:59PM. If you will need more time than this to complete your revisions, please reply to this message or contact the journal office at plosone@plos.org . Please include the following items when submitting your revised manuscript:

We look forward to receiving your revised manuscript.

Kind regards,

Mehran Rahimlou, PhD

Academic Editor

PLOS ONE

Journal Requirements:

[This work is supported, in whole or in part, by the Bill & Melinda Gates Foundation and University of Copenhagen. Under the grant conditions of the Bill & Melinda Gates Foundation, a Creative Commons Attribution 4.0 Generic License has already been assigned to the Author Accepted Manuscript version that might arise from this submission. USDA is an equal opportunity employer and provider].

4. In the online submission form, you indicated that [Data cannot be shared publicly due to concerns regarding participant/patient anonymity. Requests to access the datasets should be directed to the corresponding author].

5. We note that Figure 1 in your submission contain copyrighted images. All PLOS content is published under the Creative Commons Attribution License (CC BY 4.0), which means that the manuscript, images, and Supporting Information files will be freely available online, and any third party is permitted to access, download, copy, distribute, and use these materials in any way, even commercially, with proper attribution. For more information, see our copyright guidelines: http://journals.plos.org/plosone/s/licenses-and-copyright .

We recommend that you contact the original copyright holder with the Content Permission Form (http://journals.plos.org/plosone/s/file?id=7c09/content-permission-form.pdf ) and the following text:

“I request permission for the open-access journal PLOS ONE to publish XXX under the Creative Commons Attribution License (CCAL) CC BY 4.0 (http://creativecommons.org/licenses/by/4.0/ ). Please be aware that this license allows unrestricted use and distribution, even commercially, by third parties. Please reply and provide explicit written permission to publish XXX under a CC BY license and complete the attached form.”

Reviewers' comments:

Reviewer's Responses to Questions

**Comments to the Author**

1. Is the manuscript technically sound, and do the data support the conclusions?

Reviewer #1: Yes

Reviewer #2: Yes

Reviewer #3: Yes

Reviewer #4: Yes

Reviewer #5: Partly

2. Has the statistical analysis been performed appropriately and rigorously? 

Reviewer #1: Yes

Reviewer #2: Yes

Reviewer #3: Yes

Reviewer #4: Yes

Reviewer #5: Yes

3. Have the authors made all data underlying the findings in their manuscript fully available?

Reviewer #1: Yes

Reviewer #2: Yes

Reviewer #3: Yes

Reviewer #4: No

Reviewer #5: Yes

4. Is the manuscript presented in an intelligible fashion and written in standard English?

Reviewer #1: Yes

Reviewer #2: Yes

Reviewer #3: No

Reviewer #4: Yes

Reviewer #5: Yes

5. Review Comments to the Author

Reviewer #1: Actually it's fantastic work, but you have long abstract if it short will be great papnd my comment

Clarity and Structure: Simplify complex sentences and provide a concise summary of the methodology.

Scientific Rigor: Clarify the rationale for marker selection and indicate if multiple comparison adjustments were made.

Results Presentation: Use tables or figures to enhance accessibility and specify adjustments for comparisons.

Findings Interpretation: Expand on mechanisms linking HM composition to growth, especially in The Gambia, while addressing potential confounders.

Ethical Considerations: Explicitly state ethical approval and informed consent procedures.

Language and Style: Revise grammatical issues and lengthy sentences for improved readability.

Public Health Implications: Highlight how findings could inform interventions in resource-poor settings.

Limitations: Mention limitations like selection bias or uncontrolled confounders.

Reviewer #2: Review Comments to the Author

General Comments:

The study titled "Cytokines and appetite-regulating hormones in human milk and associations with infant growth across four sites in a longitudinal cohort" investigates important and novel aspects of maternal and infant health. The multi-site design and extensive data collection add significant value to the field. However, there are areas where the manuscript can be improved to enhance clarity, interpretability, and scientific rigor.

Strengths:

Relevance and Novelty: The focus on cytokines and appetite-regulating hormones in human milk across diverse populations is novel and addresses a significant research gap.

Design and Scope: The multi-country longitudinal design adds robustness to the findings and allows for meaningful comparisons.

Detailed Analysis: The use of ANCOVA and linear regression models provides a comprehensive assessment of associations between milk markers and infant growth.

Major Comments:

Clarity in Methodology:

While the manuscript describes sample collection and analytical methods in detail, additional clarification on how non-detectable values were handled across all sites is needed. For example, were ND values imputed consistently across all markers?

The description of adjustments made for batch variability and inter-assay variability could be expanded to include specific steps to ensure reproducibility.

Statistical Analysis:

The use of log-transformed data for normality and subsequent back-transformation for reporting results is appropriate. However, it would help if the authors justified why specific covariates (e.g., maternal BMI) were chosen over others in the ANCOVA models.

Sensitivity analyses stratified by SCM status were mentioned but not extensively discussed in the results. Including a supplementary table summarizing key findings from this stratification could improve the manuscript.

Discussion and Interpretation:

The interpretation of cytokine levels differing across countries could be expanded to include potential cultural, dietary, and genetic influences.

The associations between human milk markers and infant growth in The Gambia, but not in other sites, are interesting but require a more in-depth discussion regarding possible underlying mechanisms.

Data Availability and Ethics:

The authors state that data are restricted due to confidentiality. Consider specifying how researchers can request access and what criteria will be used to evaluate such requests.

The manuscript adheres to ethical standards, but more detail on how local differences in ethical approvals were harmonized across sites could be provided.

Minor Comments:

Abstract:

The conclusion in the abstract mentions “atopic diseases” but does not clearly connect this to the study findings. Consider rephrasing to highlight only findings directly supported by the data.

Figures and Tables:

Figure legends could benefit from additional details to ensure standalone interpretability.

Table 3 presents adjusted means but does not specify confidence intervals consistently.

Language and Grammar:

Minor grammatical errors are present, such as “The results likely reflect, that different environmental exposures...” which could be rephrased for clarity.

Conclusion:

The manuscript is a valuable contribution to understanding the interplay between human milk composition and infant growth in diverse settings. However, addressing the above concerns will strengthen the rigor and clarity of the paper.

Reviewer #3: The manuscript provides a well-conducted piece of scientific research, with clear and reliable data that supports the conclusions. The experiments are carried out with careful attention to detail, ensuring that proper controls, replication, and sample sizes are in place. The conclusions was been writing based on the data in a thoughtful and logical way, accurately reflecting the findings of the research.

The statistical analysis appears to have been performed appropriately and rigorously R soft wears. However, it might be beneficial to consider increasing the cut of point of 1.5 or 2. This adjustment could help reduce the exclusion of significance certain variables, potentially leading to a more balanced and interpretable model.

Its good data in the manuscript or included as supporting information, or alternatively. In addition to summary, because it is correct statistics raw data points underlying the means, medians, and variance measures are available. This allows for greater transparency and enables other researchers to verify and build upon the findings.

Reviewer #4: Well-written and interesting paper.

The authors collected information on socio-economic status ob 3 vists by the mothers. However, it is not presented in any table or figure. Please include this in the manuscript, if this information was collected. As socioeconomic conditions were mentioned as possible reasons for the significant findings from Gambia, analysis of this variable may be important. The authors to conisder this.

Reviewer #5: Major Comments

1. Study Design and Methodology

Clarity of Objectives: The objectives of the study are clearly stated, focusing on cytokine and appetite-regulating hormone concentrations in human milk and their association with infant growth. However, further elaboration on how these objectives align with existing literature would strengthen the introduction.

Sample Size Justification: While the manuscript mentions a sample size of 825 mother-infant dyads, it lacks a detailed justification for this size in terms of statistical power. Including a power analysis would enhance the credibility of the findings.

2. Data Collection and Analysis

Cytokine Measurement Techniques: The methods used for measuring cytokines and appetite-regulating hormones should be described in greater detail, including any standardization processes or validation methods that were employed.

Statistical Analysis: The use of analysis of covariance (ANCOVA) is appropriate; however, the manuscript should specify which covariates were included in the model. Additionally, it would be beneficial to discuss how potential confounding factors were controlled.

3. Results Interpretation

Geometric Means Reporting: The results section reports geometric means for cytokines and hormones, but it could benefit from including confidence intervals or standard deviations to provide a clearer picture of variability.

Contextualization of Findings: The differences observed among study sites are significant; however, the discussion should better contextualize these findings within existing research on environmental factors affecting human milk composition.

4. Infant Growth Associations

Z-score Calculations: The method for calculating Z-scores for infant growth should be explicitly detailed to ensure reproducibility. It would also be helpful to clarify how these scores were interpreted in relation to growth standards.

Causal Inferences: While associations between HM markers and infant growth are noted, caution should be exercised in making causal inferences without longitudinal data supporting these claims.

5. Ethical Considerations

Ethics Statement Clarity: The ethics statement is comprehensive but could benefit from a more concise summary of ethical approvals obtained across different sites to enhance readability.

6. Funding and Competing Interests

Transparency in Funding Sources: The funding disclosure is provided; however, it should explicitly state whether funders had any role in study design or data interpretation to maintain transparency.

7. Conclusion Strengthening

Implications of Findings: The conclusion summarizes the results effectively but could further emphasize the implications for public health or future research directions based on the findings regarding human milk composition and infant growth.

6. PLOS authors have the option to publish the peer review history of their article (what does this mean? ). If published, this will include your full peer review and any attached files.

**Do you want your identity to be public for this peer review?** For information about this choice, including consent withdrawal, please see our Privacy Policy .

Reviewer #1: **Yes: ** Hussein Mussa Muafa

Reviewer #2: **Yes: ** Mahdy Ali Ahmad Osman

Reviewer #3: No

Reviewer #4: No

Reviewer #5: No

---

## [Author Response · Author response to Decision Letter 0]

26 Mar 2025

Thank you very much for your valuable comments, which improved our manuscript. We have answered your comments below and addressed them appropriately in the revised manuscript. We believe the manuscript has been improved and hope it is satisfying for the reviewers.

Journal Requirements:

We will check and correct this, when resubmitting our manuscript. The correct statement is described in the Funding Information in the revised manuscript.

Grant number has now been added to the Funding Information section of the manuscript (line 748).

[This work is supported, in whole or in part, by the Bill & Melinda Gates Foundation and University of Copenhagen. Under the grant conditions of the Bill & Melinda Gates Foundation, a Creative Commons Attribution 4.0 Generic License has already been assigned to the Author Accepted Manuscript version that might arise from this submission. USDA is an equal opportunity employer and provider].

This have now been added (line 751-752)

This has been added.

4. In the online submission form, you indicated that [Data cannot be shared publicly due to concerns regarding participant/patient anonymity. Requests to access the datasets should be directed to the corresponding author].

We understand. The Data Availability Statement has been changed to: “Data cannot be shared publicly due to concerns regarding participant/patient anonymity. All data underlying the findings are to be found in the manuscript.”

5. We note that Figure 1 in your submission contain copyrighted images. All PLOS content is published under the Creative Commons Attribution License (CC BY 4.0), which means that the manuscript, images, and Supporting Information files will be freely available online, and any third party is permitted to access, download, copy, distribute, and use these materials in any way, even commercially, with proper attribution. For more information, see our copyright guidelines: http://journals.plos.org/plosone/s/licenses-and-copyright.

Figure 1 is made by the author, i.e., the original copyright holder. We assume that we should not do anything regarding this, but let us know if this is not the case.

The reference list has been checked.

Reviewers' comments:

Reviewer #1: Actually it's fantastic work, but you have long abstract if it short will be great papnd my comment

Thank you, we have shortened the abstract. I assume the comments below regard to the whole manuscript and not only the abstract, but please, let me know if this is the case.

Clarity and Structure: Simplify complex sentences and provide a concise summary of the methodology. This has been attempted (e.g., line 43-49 and 58-62).

Scientific Rigor: Clarify the rationale for marker selection and indicate if multiple comparison adjustments were made. This have now been added to line 175-177. Adjustment for multiple comparison has been indicated by the use of Bonferroni adjustments e.g., in line 236 and line 237.

Results Presentation: Use tables or figures to enhance accessibility and specify adjustments for comparisons. We are not entirely sure of what the reviewer is suggesting, but the choice of adjusting factors (possible confounding factors) were chosen based on “current evidence and biologically plausible explanations“ specified in line 226-227. The explanation is that the study is explorative, and we did not want to exclude any possible confounding by only choosing confounders based on present data. For pairwise comparisons, we did not include any confounders, which has now been added (line 238). We hope this is sufficient.

Findings Interpretation: Expand on mechanisms linking HM composition to growth, especially in The Gambia, while addressing potential confounders. This has been elucidated in line 439-440 and 443-446.

Ethical Considerations: Explicitly state ethical approval and informed consent procedures. This has been addressed in line 108-113, however, informed consent procedures differed slightly among the four study sites to accommodate cultural and practical differences.

Language and Style: Revise grammatical issues and lengthy sentences for improved readability. This has been attempted.

Public Health Implications: Highlight how findings could inform interventions in resource-poor settings. This has been elucidated in line 526-529.

Limitations: Mention limitations like selection bias or uncontrolled confounders. This has been addressed in line 503-507.

Reviewer #2: Review Comments to the Author

General Comments:

The study titled "Cytokines and appetite-regulating hormones in human milk and associations with infant growth across four sites in a longitudinal cohort" investigates important and novel aspects of maternal and infant health. The multi-site design and extensive data collection add significant value to the field. However, there are areas where the manuscript can be improved to enhance clarity, interpretability, and scientific rigor.

Strengths:

Relevance and Novelty: The focus on cytokines and appetite-regulating hormones in human milk across diverse populations is novel and addresses a significant research gap.

Design and Scope: The multi-country longitudinal design adds robustness to the findings and allows for meaningful comparisons.

Detailed Analysis: The use of ANCOVA and linear regression models provides a comprehensive assessment of associations between milk markers and infant growth.

Thank you very much for acknowledging these aspects.

Major Comments:

Clarity in Methodology:

While the manuscript describes sample collection and analytical methods in detail, additional clarification on how non-detectable values were handled across all sites is needed. For example, were ND values imputed consistently across all markers? This is stated and now elaborated in line 181-182.

The description of adjustments made for batch variability and inter-assay variability could be expanded to include specific steps to ensure reproducibility. This is elaborated in line 183-187.

Statistical Analysis:

The use of log-transformed data for normality and subsequent back-transformation for reporting results is appropriate. However, it would help if the authors justified why specific covariates (e.g., maternal BMI) were chosen over others in the ANCOVA models. Confounding factors were chosen based on “current evidence and biologically plausible explanations” as the study is exploratory (line 226-227).

Sensitivity analyses stratified by SCM status were mentioned but not extensively discussed in the results. Including a supplementary table summarizing key findings from this stratification could improve the manuscript. A supplementary table is in fact submitted as Supplementary Table S3 and the results have been discussed in line 383-387.

Discussion and Interpretation:

The interpretation of cytokine levels differing across countries could be expanded to include potential cultural, dietary, and genetic influences. We agree and have elaborated on this in line 399-410.

The associations between human milk markers and infant growth in The Gambia, but not in other sites, are interesting but require a more in-depth discussion regarding possible underlying mechanisms. We agree and have emphasized the potential mechanisms in line 439-440 and 443-446.

Data Availability and Ethics:

The authors state that data are restricted due to confidentiality. Consider specifying how researchers can request access and what criteria will be used to evaluate such requests. Thank you for highlighting this, this has been rephrased as all data is available in the manuscript.

The manuscript adheres to ethical standards, but more detail on how local differences in ethical approvals were harmonized across sites could be provided.

Minor Comments:

Abstract:

The conclusion in the abstract mentions “atopic diseases” but does not clearly connect this to the study findings. Consider rephrasing to highlight only findings directly supported by the data. We agree and have rephrased (line 58-62).

Figures and Tables:

Figure legends could benefit from additional details to ensure standalone interpretability. This has been attempted.

Table 3 presents adjusted means but does not specify confidence intervals consistently. We are not sure about this comment, as Table 3 presents 95% confidence interval for adjusted means for each of the four study sites (this is highlighted as “covariate-adjusted mean [95% CI]” described in the first line under each site). This has been further described in the table text. The right column presents the F-statistics with degrees of freedom for between and within site differences, respectively, in parentheses. We hope this is sufficient.

Language and Grammar:

Minor grammatical errors are present, such as “The results likely reflect, that different environmental exposures...” which could be rephrased for clarity. This has been attempted, thank you.

Conclusion:

The manuscript is a valuable contribution to understanding the interplay between human milk composition and infant growth in diverse settings. However, addressing the above concerns will strengthen the rigor and clarity of the paper.

Thank you for acknowledging our work and we hope we have addressed the suggestions sufficiently. We believe the manuscript has been improved markedly.

Reviewer #3: The manuscript provides a well-conducted piece of scientific research, with clear and reliable data that supports the conclusions. The experiments are carried out with careful attention to detail, ensuring that proper controls, replication, and sample sizes are in place. The conclusions was been writing based on the data in a thoughtful and logical way, accurately reflecting the findings of the research.

Thank you very much for acknowledging our work

The statistical analysis appears to have been performed appropriately and rigorously R soft wears. However, it might be beneficial to consider increasing the cut of point of 1.5 or 2. This adjustment could help reduce the exclusion of significance certain variables, potentially leading to a more balanced and interpretable model. We are not quite sure about what the reviewers refers to here. What cut off points do the reviewer suggests to increase?

Its good data in the manuscript or included as supporting information, or alternatively. In addition to summary, because it is correct statistics raw data points underlying the means, medians, and variance measures are available. This allows for greater transparency and enables other researchers to verify and build upon the findings. The raw data (median concentrations) is presented in Table 2, which we hope is sufficient.

Reviewer #4: Well-written and interesting paper.

The authors collected information on socio-economic status ob 3 vists by the mothers. However, it is not presented in any table or figure. Please include this in the manuscript, if this information was collected. As socioeconomic conditions were mentioned as possible reasons for the significant findings from Gambia, analysis of this variable may be important. The authors to conisder this. We fully agree that the socio-economic parameters could partly explain some of the findings (mentioned in line 443-446). However, these parameters have not been included for two reasons. Firstly (and importantly), the mother-infant dyads were recruited based on the same very strict inclusion and exclusion criteria ensuring that women were healthy, eating a varied diet (and had the finances to do so), living in urban areas etc. This resulted in a lower variation in socio-economic parameters, which we therefore believe are less important for the results. This has now been emphasized in the manuscript (line 446-448 – and has also been commented on in line 503-507 in relation to selection bias). Secondly, the socio-economic parameters were collected slightly differentl

---

## [Editor Report · Decision Letter 1]

4 Apr 2025

Cytokines and appetite-regulating hormones in human milk and associations with infant growth across four sites in a longitudinal cohort: The Mothers, Infants and Lactation Quality Study

PONE-D-24-37315R1

Dear Dr. Sophie Hilario Christensen,

We’re pleased to inform you that your manuscript has been judged scientifically suitable for publication and will be formally accepted for publication once it meets all outstanding technical requirements.

Kind regards,

Mehran Rahimlou, PhD

Academic Editor

PLOS ONE
---

## [Editor Report · Acceptance letter]

PONE-D-24-37315R1

PLOS ONE

Dear Dr. Christensen,

I'm pleased to inform you that your manuscript has been deemed suitable for publication in PLOS ONE. Congratulations! Your manuscript is now being handed over to our production team.

Kind regards,

on behalf of

Dr. Mehran Rahimlou

Academic Editor

PLOS ONE